# Developing an evaluation framework for public health environmental surveillance: Protocol for an international, multidisciplinary Delphi consensus study

Douglas G. Manuel[1,2,3,4]*, Carol Bennett[1,2], Emma Brown[1], David L. Buckeridge[5], Yoni Freedhoff[2], Sarah Funnell[6], Farah Ishtiaq[7], Matthew J. Wade[8,9], David Moher[3,10], on behalf of the PHES-EF Executive Group¶

1 Methodological and Implementation Research Program, Ottawa Hospital Research Institute, Ottawa, Ontario, Canada, 2 Department of Family Medicine, University of Ottawa, Ottawa, Ontario, Canada, 3 School of Epidemiology and Public Health, Faculty of Medicine, University of Ottawa, Ottawa, Ontario, Canada, 4 C.T. Lamont Primary Health Care Research Centre Program, Bruyère Research Institute, Ottawa, Ontario, Canada, 5 School of Population and Global Health, McGill University, Montreal, Quebec, Canada, 6 Department of Family Medicine, Faculty of Health Sciences, Queen's University, Kingston, Ontario, Canada, 7 Tata Institute for Genetics and Society, National Centre for Biological Sciences, Bengaluru, Karnataka, India, 8 Data Analytics and Surveillance Group, UK Health Security Agency, London, United Kingdom, 9 School of Engineering, Newcastle University, Newcastle-upon-Tyne, United Kingdom, 10 Centre for Journalology, Methodological and Implementation Research Program, Ottawa Hospital Research Institute, Ottawa, Ontario, Canada

¶ Membership of the PHES-EF Executive Group is provided in the Acknowledgments.
* dmanuel@ohri.ca

## Abstract

### Introduction

Public health environmental surveillance has evolved, especially during the coronavirus pandemic, with wastewater-based surveillance being a prominent example. As surveillance methods expand, it is important to have a robust evaluation of surveillance systems. This consensus study will develop an evaluation framework for public health environmental surveillance, informed by the expanding practice of wastewater-based surveillance during the pandemic.

### Methods

The public health environmental surveillance evaluation framework will be developed in five steps. In Step 1, a multinational and multidisciplinary executive group will be formed to guide the framework development process. In Step 2, candidate items will be generated by conducting relevant scoping reviews and consultation with the study executive group. In Step 3, an international electronic Delphi will be conducted over two rounds to develop consensus on items for the framework. In Step 4, the executive group will reconvene to finalize the evaluation framework, discuss standout items, and determine the dissemination strategies. Lastly, Step 5 will focus on

**Data availability statement:** No datasets were generated or analysed during the current study. All relevant data from this study will be made available upon study completion.

**Funding:** This project is funded by the Canadian Institutes of Health Research-funded network, CoVaRR-Net (Coronavirus Variants Rapid Response Network. FRN: 175622), and Health Canada (through the Safe Restart Agreement Contribution Program. Arrangement #: 2223-HQ-000098). The Canadian Institutes of Health Research and Health Canada have not been involved in the design or conduct of the study and the views expressed herein do not necessarily represent the views of either funding organization.

**Competing interests:** The authors have declared that no competing interests exist.

disseminating the evaluation framework to all parties involved with or affected by wastewater-based surveillance using traditional and public-oriented methods.

## Discussion

The Delphi consensus study will provide multidisciplinary and multinational consensus for the evaluation framework, by providing a set of minimum criteria required for the evaluation of public health environmental surveillance systems. The evaluation framework is intended to support the sustainability of environmental surveillance and improve its implementation, reliability, credibility, and value.

## Introduction

Public health relies on effective surveillance systems to support informed decision-making and timely interventions [1]. Wastewater-based surveillance, a form of environmental-based surveillance, has historically informed public health responses to threats like typhoid, polio, and illicit drug use [2–6]. More recently, the coronavirus pandemic (Covid-19) marked a turning point, expanding wastewater-based monitoring and surveillance to an unprecedented scale and underscoring its potential for wider public health applications [7]. Since then, wastewater-based surveillance, has supported public health practice for controlling monkeypox [8], influenza [9,10], antimicrobial resistance [11,12], and other pathogens and health risks beyond infectious diseases [13,14]. It is now used globally across diverse settings [15], from metropolitan to rural areas [16,17]. Cited advantages include its ability to track disease transmission independently of clinical testing availability or health-seeking behaviour, offering a non-invasive, resource-efficient surveillance method [18,19].

Current frameworks for evaluating surveillance performance require reassessment and updating. Established guidelines, such as the U.S. Centers for Disease Control and Prevention (CDC) 2001 framework, were developed before widespread environmental-based surveillance and do not adequately address unique challenges like environmental measurement variability or the need for broad, multidisciplinary collaboration [5,20,21]. Meanwhile, the pandemic prompted shifts in surveillance practices—such as increased public engagement and open-access data—and encouraged consensus-driven guidance through international, multidisciplinary collaboration. Yet no systematic review has examined how these developments influence surveillance evaluation, and there is no rigorous consensus on measuring the performance of wastewater-based surveillance and other forms of environmental-based surveillance.

This study aims to develop a comprehensive public health evaluation framework. Wastewater-based surveillance will be an example; however, it will apply to broader public health environmental surveillance systems, including air and surface monitoring. Surveillance targets considered include infectious agents and health risks beyond SARS-CoV-2.

## Study objectives

This project will use a multidisciplinary and multinational consensus approach to develop an evaluation framework for public health environmental surveillance (**P**ublic **H**ealth **E**nvironmental **S**urveillance **E**valuation **F**ramework, PHES-EF). See https://PHES-EF.org. An evaluation framework provides structured guidance to facilitate a systematic approach to program evaluation [22–24]. See below for the study's definition of an evaluation framework.

Specific objectives for the framework development process are as follows: (1) engage a multidisciplinary executive group of international experts to inform the development of the evaluation framework; (2) review the literature to identify current guidance on surveillance evaluation items; (3) prioritize items for an evaluation framework for public health environmental surveillance using an international electronic-Delphi (e-Delphi) process; (4) conduct a consensus meeting, which engages experts across disciplines, to create an evaluation framework for public health environmental surveillance based on feedback from the international e-Delphi process; and, (5) develop a comprehensive dissemination plan for the evaluation framework.

## Definitions

We define key terms as follows:

- **Environmental-based surveillance (environmental surveillance):** Surveillance of substances or organisms in air, water, soil, and living organisms to detect changes or threats that may affect human health, ecosystems, or both [25].

- **Evaluation framework:** A structured system or schema to assess a program systematically [1,22,24]. An evaluation framework presents a clear list of items or elements that should appear in an evaluation, such as constructs or concepts. It provides an organisation or structure for the list of items, defines how the list and organisation were developed, and potentially includes details about measuring the evaluation items. Importantly, an evaluation framework facilitates informed decision-making by providing a comprehensive understanding of program effectiveness and areas for improvement.

- **Public health surveillance:** The ongoing systematic collection, analysis, interpretation and dissemination of health data for the planning, implementation and evaluation of public health interventions and strategies to protect and improve the health of populations [26].

- **Wastewater:** Water that has been in contact with people (e.g., for washing) or used for cleansing and sanitation (e.g., for flushing away faecal matter), and is discharged via sewers or other sanitation systems [6].

- **Wastewater-based surveillance (wastewater surveillance):** A public health approach that examines specific substances or organisms in wastewater (sewage). Wastewater surveillance can be referred to as wastewater-based epidemiology; however, the latter can be specific to using wastewater to study and analyse the distribution, patterns, and determinants of health and disease conditions in defined populations [26].

# Materials and methods

## Study design

This protocol was developed *a priori*. The development of PHES-EF is guided by recommendations for Delphi techniques in the health sciences [27]. Fig 1 provides an overview of the study flow.

## Step 1: Establish the executive group

A multinational and multidisciplinary executive group will oversee the development of the evaluation framework. The group will bring expertise from various fields, including public health, infectious disease, epidemiology, environmental and physical sciences, mathematical sciences, social sciences, communication, knowledge translation and exchange, knowledge users, and an engaged public.

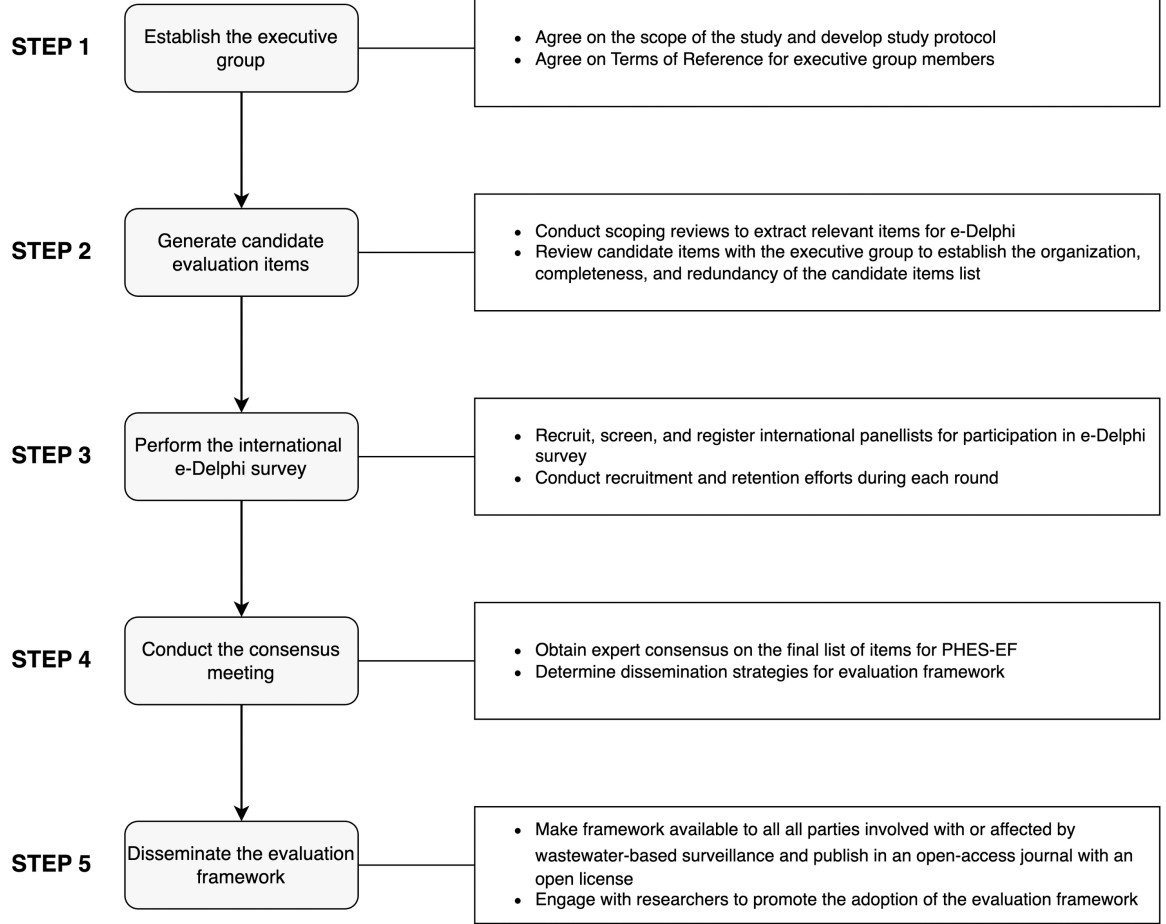

| STEP 1 | Establish the executive group | • Agree on the scope of the study and develop study protocol<br>• Agree on Terms of Reference for executive group members |
|---|---|---|
| STEP 2 | Generate candidate evaluation items | • Conduct scoping reviews to extract relevant items for e-Delphi<br>• Review candidate items with the executive group to establish the organization, completeness, and redundancy of the candidate items list |
| STEP 3 | Perform the international e-Delphi survey | • Recruit, screen, and register international panellists for participation in e-Delphi survey<br>• Conduct recruitment and retention efforts during each round |
| STEP 4 | Conduct the consensus meeting | • Obtain expert consensus on the final list of items for PHES-EF<br>• Determine dissemination strategies for evaluation framework |
| STEP 5 | Disseminate the evaluation framework | • Make framework available to all all parties involved with or affected by wastewater-based surveillance and publish in an open-access journal with an open license<br>• Engage with researchers to promote the adoption of the evaluation framework |

**Fig 1. Overview of the development process for PHES-EF.**

The composition of the study executive group, which includes content expert and knowledge user members, will be identified by examining relevant author lists, internet-based searching, and the study working group's professional networks. In contrast, the engaged public member of the executive group will be identified through an open call for expressions of interest. The S1 Table details the composition of the study executive group.

The executive group will be responsible for providing feedback on the framework development process (e.g., framework organization, completeness, additions, redundancy), supporting and advising working groups in their area of expertise, connecting the study to other related projects, recruiting e-Delphi panellists, participating in the consensus meeting, disseminating the evaluation framework, advising and/or participating in post-publication activities, and promoting the adoption of the evaluation framework among their professional networks. Co-chairs will also be responsible for decision-making when the executive group cannot reach a consensus, leading executive group activities, and the overall framework development process.

## Step 2: Generate candidate evaluation items

We will systematically compile a list of e-Delphi candidate items through scoping reviews, content analysis and extraction, and executive group consultation.

## Scoping reviews

Two scoping reviews will be performed to provide a comprehensive view of public health environmental surveillance system evaluations. The first review will identify frameworks (e.g., sources which guide the planning, management, or conduct of processes) and criteria (e.g., principles on which a judgment may be based) for evaluating public health or environmental surveillance systems. The second review will explore insights from the Covid-19 pandemic, particularly focusing on the principles of equity, trust, open science, community engagement, and One Health in the context of waste-water surveillance [28–30]. Detailed protocols for the scoping reviews are available on OSF (https://doi.org/10.17605/OSF.IO/GSKF6; https://doi.org/10.17605/OSF.IO/C9A3Q). The completed scoping reviews will be reported following the Preferred Reporting Items for Systematic Reviews and Meta-Analyses extension for scoping reviews (PRISMA-ScR) [31]. See S1 and S2 Appendices for associated PRISMA-ScR checklists.

## Content analysis and extraction

We will conduct a content analysis on items identified during the data extraction phase for both scoping reviews. The content analyses will identify and group criteria for evaluating public health and environmental surveillance systems. In the first step, one team member will group all the extracted items into initial domains and themes. In the second step, two researchers will independently compare the category system of the first researcher to agree on the final category system consensually. In the third step, all criteria will be reviewed again to merge or modify duplicate concepts and create a parsi-monious framework for the initial list of e-Delphi candidate items.

## Executive group consultation

The executive group will review the initial list of e-Delphi candidate items and discuss the initial structure and presenta-tion of Round 1 items. Their review will focus on creating an evidence-informed public health environmental surveillance framework centred around SARS-CoV-2 wastewater surveillance, and incorporating lessons learned from the pandemic (e.g., equity, open science, community engagement, and One Health). Additional items will be added if the executive group determines that key evaluation areas have not been addressed.

## Step 3: International e-Delphi survey

A Delphi technique will be used to develop an internationally accepted evaluation framework for public health environ-mental surveillance. The Delphi method is an iterative multi-round approach that uses a series of sequential surveys, interspersed by controlled feedback, to elicit consensus among a group of individuals while maintaining anonymity [32,33]. An e-Delphi method will be used to overcome geographic barriers and allow us to engage panellists internationally across various time zones. Delphi surveys relating to health sciences have typically been conducted for 2–3 rounds [27,34,35]. Therefore, a two-round e-Delphi survey and a subsequent consensus meeting will be conducted to balance feasibility with rigour. Each round will be active for approximately 4–6 weeks to allow panellists sufficient time to participate in the survey while balancing study timeline constraints [36].

## Recruitment of panellists

We will recruit a multinational, multidisciplinary panel of wastewater surveillance experts, knowledge users, and engaged members of the public to complete the e-Delphi survey. Content experts are adult individuals with knowledge in their respective subdiscipline groups (see S2 Table). Panellists will be selected to capture the multiple perspectives of those that influence the design, implementation, evaluation, use, and reporting of wastewater surveillance activities, includ-ing the following subdisciplines groups: public health, infectious disease, epidemiology; environmental and physical sciences; mathematical sciences; social sciences; and communication, knowledge translation and exchange. Targeted

recruitment will also be conducted to identify knowledge users and engaged members of the public. We will use purposive sampling to identify potentially eligible panellists by examining relevant author lists from the scoping reviews, professional networks, and internet-based searching, among other potential methods. Potential panellists may also be recruited using social media, a study website or calls for expressions of interest during relevant meetings, presentations, or correspondence. Snowball sampling may also be employed by executive group members and potential panellists to maximize recruitment.

Potentially eligible panellists will receive a recruitment email from the study working group, executive group members, or other potential panellists. A call for panellists to share the e-Delphi with other potentially eligible panellists may be included in recruitment materials and the e-Delphi survey. Inclusion criteria for e-Delphi panellists are outlined in Table 1.

We will conduct an all-rounds e-Delphi invitation approach, meaning panellists will be invited to every round regardless of whether they participated in the previous round [37]. Previous Delphi studies have commonly ranged from 20 to 30 panellists; however, there is no agreed upon standard [35,38]. Studies have also shown that the number of panellists and their diversity can impact the quality of outcomes [38]. Therefore, assuming only half of registered panellists will participate in at least one round, we will aim to recruit at least 50 panellists. Preferably, with at least 8–10 people per discipline subgroup. The study working group will monitor the distribution of registered panellists based on their demographic information and will try to distribute appropriately across discipline subgroups and other demographic markers.

Registration to complete the e-Delphi will open approximately four weeks before the first round, and e-Delphi round surveys will be active for roughly 4–6 weeks. The tentative start and end dates for recruitment are 08/09/2025 and 19/12/2025, respectively. Recruitment or retention efforts will occur throughout each e-Delphi round and the four weeks of targeted recruitment before Round 1. Email reminders to complete the survey will be sent twice per e-Delphi round. Panellists may also receive up to two reminders to complete the screening questionnaire during the targeted recruitment period. The time required to complete each e-Delphi round survey will be approximately one hour. The estimated time commitment will be provided to all potential panellists during recruitment and before starting the consensus process.

### Panellist withdrawal or termination

Panellists will not be able to withdraw from the study due to the safeguards in place to maintain anonymity; however, they may choose to stop actively participating in the study at any time.

Table 1. Eligibility criteria for e-Delphi survey panellists.

| Discipline group | Discipline subgroup | Inclusion criteria |
|---|---|---|
| Content experts | Public health, infectious disease, epidemiology<br>Environmental and physical sciences<br>Mathematical sciences<br>Social sciences<br>Communication, knowledge translation and exchange | An adult[a] who is proficient in English and has a graduate degree in one of the listed specializations[b], or ≥ 3 years of professional experience, or ≥ 2 peer-reviewed publications relating to wastewater surveillance. |
| Knowledge users | | An adult[a] who is proficient in English and who is a professional who does not have specialized training or qualifications in wastewater surveillance, but who uses surveillance data to inform policy and action in their workplace. |
| Engaged public | | An adult[a] who is proficient in English and has relevant lived experience. |

Definitions: (1) Professional experience – Paid employment or professional practice in a listed specialization (current or former); (2) Graduate or professional degree – Master's or Doctor of Philosophy (PhD) relating to a listed discipline subgroup.

a Adult: ≥ 18 years of age

b See S2 Table for specializations associated with each discipline.

## Panellist compensation

Content experts and knowledge users who complete all rounds of the e-Delphi process will receive a $100 CAD pre-paid Visa card in compensation for the time required to complete the e-Delphi process. Engaged members of the public who complete all rounds will be entered to win one of five $100 CAD pre-paid Visa cards. Remuneration has been shown to encourage participation and retention from diverse groups, thereby minimizing bias, and motivating thoughtful, high-quality responses, thereby improving data quality [39].

## Procedure

We will conduct a two-round e-Delphi survey to generate consensus on evaluation criteria (see Fig 2). The full survey will be pre-tested and validated before administration. A pilot test of the survey will be conducted with up to five test panellists. Test panellists may include members of the executive group or other potential panellists matching our recruitment criteria. The pilot test will focus on assessing the clarity and relevance of background information provided for candidate items. The survey will be revised as necessary following feedback from pilot testing. Summaries of Round 1 will be compiled for the subsequent round.

We will use the Surveylet Delphi platform to administer the survey [40]. The platform facilitates interactive dialogue between panellists by allowing them to justify their responses and comment on the responses of others. Panellists can also save their progress and answer survey questions across multiple sessions. Custom survey pathways will be generated for each panellist discipline subgroup (i.e., panellists from different discipline subgroups will be shown a different collection of candidate items). Within each custom discipline subgroup survey pathway, panellists can skip items or self-declare that they are not qualified to assess certain candidate items. To ensure the independent evaluation of items, the order of candidate items may be randomized for each panellist.

Individuals will complete the screening questionnaire separate from the e-Delphi platform using LimeSurvey [41]. The study working group will review eligibility, and if eligible to participate in the study, individuals will receive an individualized link for anonymous participation in the e-Delphi survey. Implied consent will be obtained electronically for all panellists using the Surveylet Delphi platform before participation in the survey. Panellists who are eligible for compensation or wish to receive acknowledgement for their contributions will be able to do so by clicking a link at the end of each survey round. The link will bring them to a separate LimeSurvey form that will not be linked to their survey responses, allowing for e-Delphi responses to remain anonymous.

## Defining consensus

Candidate items will be assessed on the following four rating properties: (1) relevance and practical utility; (2) scientific rigor, validity and reliability; (3) feasibility, adaptability and resource implications; and (4) equity, inclusiveness, and mitigation of bias.

A 7-point Likert scale (1 = highly irrelevant to 7 = highly relevant) will be used to rate each property for all candidate items. A summary score for each candidate item will be created by calculating the median of the four property ratings. Panellist summary scores will then be categorized as excluded item (irrelevant: 1–2), further discussion (equivocal: 3–5), or core item (relevant: 6–7). Consensus for each item is defined as ≥ 70% of the panellist votes falling within the same category (1–2, 3–5, or 6–7).

The approach to use a 7-point fully-labeled Likert scale – a higher number of categories compared to many studies – is informed by the consideration that panellists are professionals and engaged members of the public with good cognitive skills; therefore, more categories and labels will be more discriminating and reproducible [42–45]. The ≥ 70% consensus cut-off – a lower cut-off compared to many studies – is informed by the rapidly evolving nature of environmental surveillance for public health and the wide range of disciplines invited to the panel; therefore, a high level of agreement may not occur [42–45].

The rating properties were synthesized from previous performance evaluation frameworks from public health and health systems [1,24,46–51]. Equity is a property that was not previously identified but was added following an expert group discussion of potential properties and the study's goals and objectives. See S3 Appendix for further details on the rating properties for e-Delphi candidate item scoring.

### Stopping rules

When consensus is reached for a given item, no subsequent ranking rounds for that item will be performed. Consensus items that are considered equivocal, in addition to remaining non-consensual items, will be further deliberated during the consensus meeting.

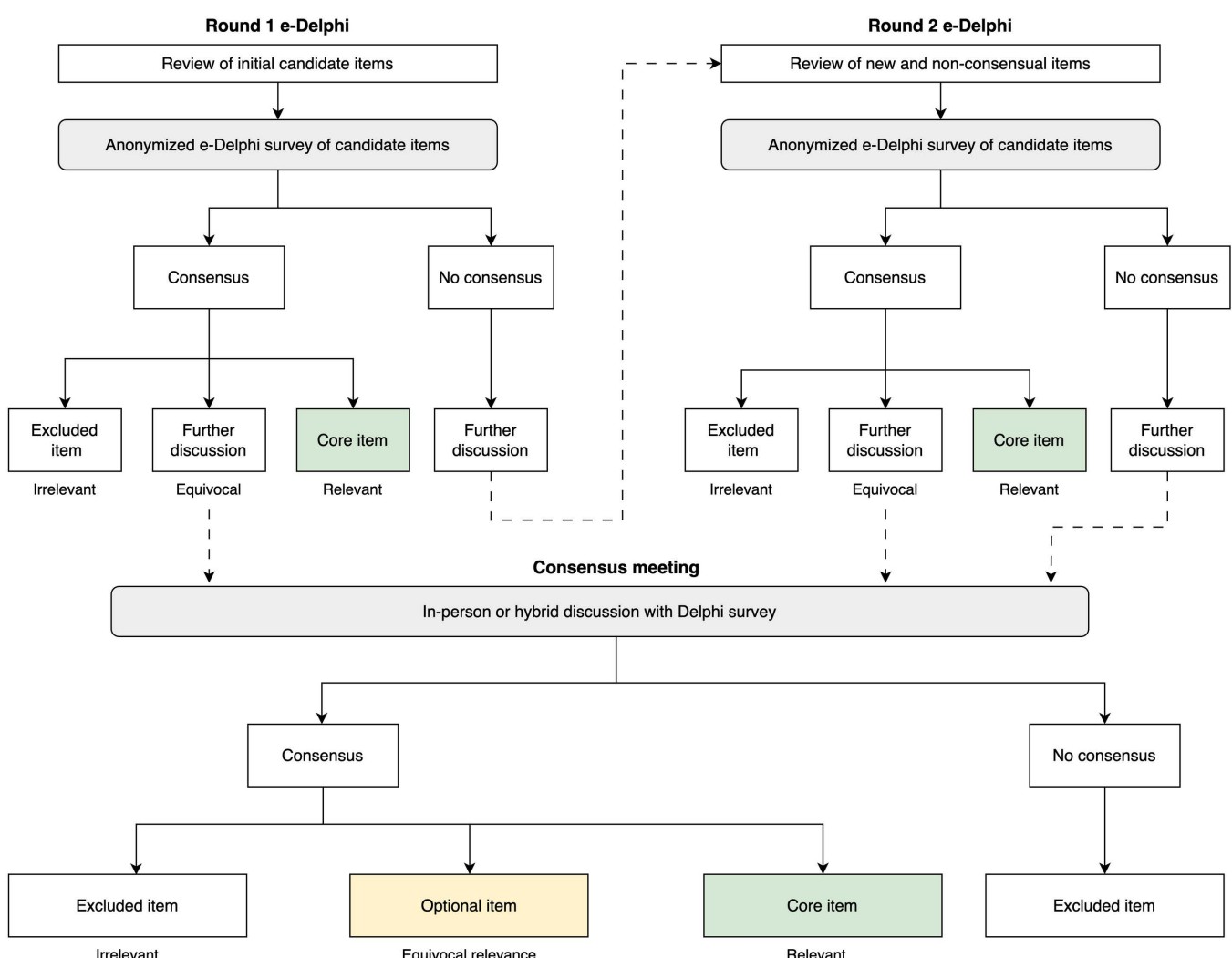

**Fig 2. E-Delphi and consensus meeting methodology, methods to achieve consensus on core items for PHES-EF.** Consensus: if ≥ 70% of panel-list votes fall within the same category (irrelevant, equivocal, relevant); no consensus if <70% agreement. Item relevancy: *core item*: highly or moderately relevant; *further discussion*: slightly relevant, neutral, slightly irrelevant; *excluded item*: moderately or highly irrelevant.

### Round 1

Panellists will be invited to rate their level of agreement with candidate items generated from scoping review results and consultation with the study executive group. Free-text boxes will be included for panellists to provide feedback or identify additional candidate items to be included in the next e-Delphi survey round.

### Round 2

Regardless of whether they participated in the previous round, panellists will be invited to participate in Round 2 of the e-Delphi survey. All panellists will be invited to rate new items and re-rate previous items that did not reach consensus. When re-rating their level of agreement, panellists will be presented with their previous round scores alongside the aggregate group results. Anonymous feedback from Round 1 will also be compiled and presented during Round 2. Any newly suggested items during Round 2 will be deliberated during the consensus meeting.

### Summary panellist characteristics

At the start of the e-Delphi survey, panellists will provide basic demographic details to ascertain their discipline subgroup, ensuring they receive the correct list of candidate items. To maintain anonymity, this demographic data will not contain identifiable information, and only aggregate responses of 5 or more panellists will be analyzed. We will report aggregate panel characteristics based on gender, discipline group or subgroup (depending on the number of panellists), level and type of experience, country income level, and World Bank work regions.

### Quantitative analysis

We will report response rates, agreement levels, consensus, and stability across rounds using median, range, and IQR for each criterion item. The proportion of panellists who skipped items or selected "not qualified to respond" will be reported. Fisher's exact test will be used to evaluate differences in item ratings by panellist characteristics: primary panellist disciplines (see S2 Table), country income level (high-income versus low-and-middle income), and World Bank world regions (see S3 Table). Fisher's exact test was chosen for its suitability in analyzing small sample sizes and categorical data, as it does not rely on the assumption of large sample sizes required by other tests, such as the chi-squared test. This makes it a good choice for comparing item ratings across subgroups, particularly given the potential variability in panellist numbers across disciplines, income levels, and regions.

### Qualitative analysis

Content analyses will be conducted on panellists' feedback to extract common themes and insights. A summary will describe the evolution of each criterion across rounds, including reasons for item modification or elimination.

### Grouping of subgroups

Subgroups with fewer than 5 panellists will be combined with similar groups based on panellist numbers and group likeness. Decisions on combined subgroups will be made prior to analyses of the items.

### Exclusion of data

All panellists that participate in at least one round will have their responses included in analyses unless the study working group determines that a panellist's response warrants being excluded from analyses for reasons that are newly developed or not previously recognized.

## Step 4: Consensus meeting

In preparation for the consensus meeting, the study working group will categorize the remaining candidate items for consideration into a preliminary evaluation framework. This will include merging and/or modifying the remaining items based on gathered feedback from the e-Delphi rounds.

We will invite all members of the executive group to participate in the consensus meeting. An in-person or hybrid (in-person and online) consensus meeting may be held using the Surveylet Delphi survey platform after the results of the e-Delphi have been compiled and analyzed. If in a hybrid format, Microsoft Teams will be used for messaging and videoconferencing. Video recording and transcription services may also be used to aid notetaking. The primary objective of the meeting will be to achieve expert consensus on the final list of items for PHES-EF, through review and discussion of salient items. This process will be guided by the empirical evidence that was identified during the scoping reviews, and the opinions gathered during the e-Delphi process. The secondary objective will be to discuss publication and dissemination strategies for the final evaluation framework.

Steps to produce the final list of items are as follows: (i) present the results of the e-Delphi exercise (name, rationale, and score of each item); (ii) discuss the rationale and relevance for including the items in the framework; and (iii) vote on equivocal and non-consensual items. PHES-EF will be developed based on the final list of items that receive consensus during the executive group consensus meeting.

## Step 5: Dissemination of evaluation framework

The dissemination of this evaluation framework will start with the publication of the PHES-EF protocol. The development of the evaluation framework will be reported in a statement document that will include the rationale and a brief description of the meeting, and the panellists involved. While the publication strategy will be finalized based on the consensus meeting discussion, the preliminary approach will be to post to OSF as a preprint, before publication in an open-access peer-reviewed journal. During this time, we will actively seek feedback from experts, knowledge users, and the public within our networks and via social media. PHES-EF will be made available using an open license (CC-BY-SA-4.0 license).

Post-publication, we will aim to keep the framework dynamic and up to date using a website, Discourse server, or other social media platforms to elicit ongoing feedback. Post-publication strategies may also include the creation of guides, toolkits, or ontologies to enhance the practical applicability of the evaluation framework and facilitate its adoption for policy and decision-making. Active dissemination approaches will include presenting at relevant scientific conferences, holding webinars, and conducting workshops. Additional dissemination practices will be determined during the consensus meeting.

## Equity, diversity, and inclusion (EDI)

Incorporating EDI into scientific research enhances creativity and innovation, and improves the quality of results by embracing diverse perspectives and minimizing biases. It also helps promote fairness and equity by aiming to ensure that research outcomes serve various communities and global needs.

"Equity is defined as the removal of systemic barriers and biases enabling all individuals to have equal opportunity to access and benefit from the program" [52]. To achieve this, members of the research team are committed to developing a strong understanding of the systemic barriers faced by individuals from underrepresented groups. "Diversity is defined as differences in race, colour, place of origin, religion, immigrant and newcomer status, ethnic origin, ability, sex, sexual orientation, gender identity, gender expression and age" [52]. Panellists will be asked to provide characteristic information (e.g., country of residence, gender, primary discipline) to assess and help ensure (through more targeted recruitment) a large diversity in perspectives. "Inclusion is defined as the practice of ensuring that all individuals are valued and respected for their contributions and are equally supported" [52]. To help ensure that all research team members are integrated and supported, we will have a code of conduct statement in the executive group Terms of Reference.

Where feasible, and with sufficient statistical power, e-Delphi results will be further analyzed by panellist demographic information, such as by distinguishing respondents from high-, middle-, and low-income settings, as well as different world regions. This will allow for a nuanced understanding of how these contexts may have influenced the development of the framework, and which candidate items may be most valued in these areas.

All study working group members will complete EDI-related Canadian Institutes of Health Research (CIHR) training [53], and the First Nations Principles of OCAP© (Ownership, Control, Access, and Possession) training for research [54]. The executive group will receive an overview of EDI and OCAP© and will be encouraged and supported to complete such training. Furthermore, staff trained in EDI and OCAP© principles will review the framework from an EDI perspective to ensure appropriate language.

### Knowledge user and public involvement

The international e-Delphi survey will collect responses from content experts, knowledge users, and engaged members of the public to ensure that all parties involved with or affected by wastewater surveillance are represented in the consensus-building approach. Apart from what has already been described, more detailed recruitment or public engagement methods may be informed by the executive group's engaged member of the public. Additionally, to ensure accurate and transparent reporting of knowledge user and public involvement throughout the study, we will refer to the Guidance for Reporting Involvement of Patients and the Public (GRIPP2) checklist [55]. We will document the methods used to engage knowledge users and members of the public, report the impacts and outcomes of their engagement, and report on lessons learned from the experience.

### Acknowledgments

E-Delphi participation will not satisfy authorship criteria; however, panellists will have the option to receive individual acknowledgement for their contributions to the framework, given they provide consent to be named. Individuals who participate in the consensus meeting will have the opportunity to be named as authors.

### Research data management

The project will follow an open-science approach and make available the data, code, and materials for all project stages, including the search strategy, search findings, initial curation of evaluation measures, summary of executive group discussions, e-Delphi processes, and agreement procedures. The management of research data will follow a data management and sharing plan.

All personal data will be managed using LimeSurvey. LimeSurvey is a free, open-source, online survey tool. Personal data will only be collected from individuals during the screening questionnaire, and from panellists who wish to receive compensation and/or individual acknowledgement for their contributions to the evaluation framework. LimeSurvey data management practices are overseen by The Ottawa Hospital (TOH) Research Institute Methods Centre that protects privacy and confidentiality while assisting authorized users in collecting and managing survey data. It has been installed on a TOH server in the TOH Demilitarized Zone and all data collected is stored in a TOH database within the TOH firewall. Because LimeSurvey is installed outside of the TOH firewall in the TOH Demilitarized Zone, surveys can be responded to from anywhere using an internet connection.

Calibrum's Surveylet Delphi platform will be used to manage e-Delphi responses. It may also be used for the consensus meeting. The e-Delphi will only collect anonymous data using a modified Delphi approach; meanwhile, the consensus meeting may collect identifiable data using a real-time Delphi approach. Surveylet complies with the EU-U.S. Privacy Shield Framework and Swiss-U.S. Privacy Shield Framework regarding the collection, use, and retention of personal information. The Surveylet platform also allows its users to be General Data Protection Regulation (GDPR) compliant. Calibrum's servers are protected by high-end firewall systems and vulnerability scans are conducted regularly. Calibrum

uses Transport Layer Security (TLS) encryption (also known as HTTPS) for all transmitted data. Surveys are also protected with passwords and HTTP referrer checking. All data are encrypted at rest and hosted by SSAE-16 SOC II certified third-party data centres. Data on deprecated hard drives are destroyed by U.S. DOD methods and delivered to a third-party data destruction service.

### Study record retention

Any data exported from either survey company will be stored securely in a password protected location within TOH's SharePoint. All study records will be retained securely for 10 years following study completion, then destroyed in accordance with Ottawa Hospital Research Institute (OHRI) requirements.

### Ethical considerations

This study has received ethical approval (20230428-01H) from the Ottawa Health Science Network Research Ethics Board (OHSN-REB). Implied consent will be obtained from all panellists prior to participating in the e-Delphi survey. Information about the survey and consent practices will be provided on the first page of the survey. Panellists will be informed that by providing their demographic information and proceeding to the next page, they are providing their implied consent to participate in the study.

Only anonymous panellist responses will be used for analyses. All panellist data and responses will be password protected and only REB-approved study working group members from the Ottawa Hospital will have access to the data.

## Discussion

This paper outlines the protocol for a two-round e-Delphi study that will be used to develop a multinational, multidisciplinary evaluation framework for public health environmental surveillance. Developing an evaluation framework will contribute to a better understanding of the ideal scope and use cases for wastewater surveillance and improve the reliability and credibility of surveillance data for public health officials. An evaluation framework will also help promote the sustainability of wastewater surveillance systems by providing an effective performance measurement tool.

We strove to design the PHES-EF protocol using a rigorous consensus approach. The development of the PHES-EF study protocol was informed by ACCORD (ACcurate COnsensus Reporting Document): A reporting guideline for consensus methods in biomedicine developed via a modified Delphi [38]. Surveillance evaluation frameworks are uncommonly developed using a consensus approach. PHES-EF will be developed for environmental surveillance and informed by wastewater surveillance during the pandemic. However, associated scoping reviews and prior surveillance evaluation frameworks are related largely to any public health surveillance system [56]. Therefore, PHES-EF may be applicable for evaluating public health surveillance beyond environmental systems.

## Limitations

Although the Delphi technique uses a systematic process to generate consensus among a panel of experts, the method is not without its limitations. One of the primary concerns when using this technique is the ability to implement a truly representative expert panel. Despite our plan to use various recruitment methods to enroll panellists from all discipline subgroups with varying demographic characteristics, consensus results may ultimately be influenced by the final panel composition [57].

Another potential limitation in using a modified Delphi approach is the potential influence of controlled feedback on the convergence of panellist opinions. Although providing controlled feedback between survey rounds aids in generating consensus, this approach may bias consensus results by influencing individual panellists' opinions [58].

Furthermore, due to feasibility and resource limitations, only sources that are available in English will be included in associated scoping reviews. Additionally, only individuals who are proficient in English will be eligible to participate in the

e-Delphi survey and consensus meeting. Therefore, the findings of this study may not reflect relevant knowledge that is not available in English.

## Supporting information

**S1 Table. Executive group composition.**
(PDF)

**S2 Table. Executive group and e-Delphi panellist disciplines.** The list of specializations or related concepts is not extensive.
(PDF)

**S3 Table. World region classifications.** Note: The regional classifications are used by the World Bank for organizing economic data, development projects, and policy analysis. These classifications are not strictly geographical but also consider economic and developmental similarities and relationships.
(PDF)

**S1 Appendix. PRISMA-ScR checklist for scoping review #1.** Current Evaluation Practices for Public Health and Environmental Surveillance Systems: A Scoping Review Protocol.
(PDF)

**S2 Appendix. PRISMA-ScR checklist for scoping review #2.** Lessons Learned from Wastewater-Based Surveillance During the COVID-19 Pandemic: A Scoping Review Protocol.
(PDF)

**S3 Appendix. Rating properties for e-Delphi candidate item scoring.**
(PDF)

**S4 Appendix. Draft e-Delphi panellist screening questionnaire.**
(PDF)

## Acknowledgments

We thank Mary Jessome (PhD candidate, University of British Columbia) for her contributions to the study as a past executive group member with expertise in social sciences. We would also like to thank Peter Farrell (Research Librarian, University of Ottawa) for developing the associated scoping review search strategies for this study.
PHES-EF Executive Group (ordered alphabetically according to surname except for the first two authors):
Douglas G. Manuel* (0000-0003-0912-0845), David L. Buckeridge* (0000-0003-1817-5047), Yoni Freedhoff (0000-0001-9391-3455), Sarah Funnell, Bernd M. Gawlik (0000-0002-4406-5362), Farah Ishtiaq (0000-0002-6762-7014), Amy E. Kirby, Kerrigan McCarthy (0000-0001-8958-9795), David Moher (0000-0003-2434-4206), Beate Sander (0000-0003-2128-9133), Jeremy Veillard, Matthew J. Wade (0000-0001-9824-7121)

* Co-chairs of PHES-EF Executive Group

## Author contributions

**Conceptualization:** Douglas G Manuel, Carol Bennett.

**Funding acquisition:** Douglas G Manuel.

**Methodology:** Douglas G Manuel, Carol Bennett, Emma Brown, David Moher.

**Supervision:** Douglas G Manuel.

**Writing – original draft:** Douglas G Manuel, Carol Bennett, Emma Brown.

**Writing – review & editing:** Douglas G Manuel, Carol Bennett, Emma Brown, David L. Buckeridge, Yoni Freedhoff, Sarah Funnell, Farah Ishtiaq, Matthew J. Wade, David Moher.

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
