## [Decision Letter · Decision Letter 0]

30 Dec 2024

PONE-D-24-33538Developing an evaluation framework for public health environmental surveillance: Protocol for an international, multidisciplinary e-Delphi studyPLOS ONE

Dear Dr. Manuel,

Thank you for submitting your manuscript to PLOS ONE. After careful consideration, we feel that it has merit but does not fully meet PLOS ONE’s publication criteria as it currently stands. Therefore, we invite you to submit a revised version of the manuscript that addresses the points raised during the review process.

The reviewers have reviewed the work you have done and recognize the contribution of the research to the field. However, they have also provided constructive comments and suggestions to improve the clarity and overall quality of the manuscript, ensuring it meets the standards for publication. Please pay particular attention to the methodology section, ensuring detailed and clear information is provided regarding the selection of panelists, the statistical methods employed, and the criteria used for reaching consensus.

We kindly request that you address these comments in a revised version of your manuscript.

We look forward to receiving your revised manuscript.

Kind regards,

Angela Mendes Freitas

Academic Editor

PLOS ONE

3. One of the noted authors is a group or consortium [PHES-EF Executive Group]. In addition to naming the author group, please list the individual authors and affiliations within this group in the acknowledgments section of your manuscript. Please also indicate clearly a lead author for this group along with a contact email address.

Reviewers' comments:

Reviewer's Responses to Questions

**Comments to the Author**

1. Does the manuscript provide a valid rationale for the proposed study, with clearly identified and justified research questions?

Reviewer #1: Yes

Reviewer #2: Yes

2. Is the protocol technically sound and planned in a manner that will lead to a meaningful outcome and allow testing the stated hypotheses?

Reviewer #1: Partly

Reviewer #2: Yes

3. Is the methodology feasible and described in sufficient detail to allow the work to be replicable?

Reviewer #1: Yes

Reviewer #2: Yes

4. Have the authors described where all data underlying the findings will be made available when the study is complete?

Reviewer #1: Yes

Reviewer #2: Yes

5. Is the manuscript presented in an intelligible fashion and written in standard English?

Reviewer #1: Yes

Reviewer #2: Yes

6. Review Comments to the Author

You may also provide optional suggestions and comments to authors that they might find helpful in planning their study.

Reviewer #1: Overall, this study protocol is well-prepared and addresses an important issue in public health surveillance. The authors’ structured approach, use of the Delphi method, and comprehensive dissemination plan indicate a strong foundation for a valuable contribution to public health. Nevertheless, to improve the study, the following items can be reviewed and revised:

• The background and rationale could be strengthened by adding a section specifically addressing the limitations of current evaluation methods and how this framework seeks to address those gaps.

• The article could include references to recent advancements or alternative applications of wastewater-based surveillance (e.g., monitoring antimicrobial resistance, drug residues).

• Consider a pilot round to identify potential issues with the survey questions or consensus-building platform (Surveylet Delphi).

• Additional details on post-publication use and integration of the framework in policy-making and public health systems would enhance practical applicability.

• How this framework might adapt to different socio-economic and geographic contexts should be discussed.

Almost, it could be say that if any other researcher were to conduct this study, they would likely use the same approach. As a result, I believe there is a lack of specific innovation or creativity in this protocol. It is recommended to consider modifications to the research methodology or interventions to enhance the protocol's appeal and impact, thereby creating more pronounced distinctions from similar studies.

Reviewer #2: The methods and protocol sections are comprehensive and well-structured. This reflects a good approach to the Delphi process. I think there is an opportunity to provide additional detail to better support clarity and detail reproducibility. For instance, specifying the scoring scale used by panelists and clearly defining the criteria for achieving consensus would strengthen the methodology. The protocol emphasizes EDI, but the selection process for panelists could be better explained to outline how diversity in expertise and demographics will be ensured, and importantly how it might shape/improve the outcomes. While the quantitative analysis is detailed, further explanation of how median, range, and IQR will be used to define consensus and stability is needed. Also, elaborating on why Fisher’s exact test was chosen and its threshold for significance would provide a stronger rationale for the approach. I like to see stats choices well defined.

The ethical considerations part is strong, though additional information on how informed consent will be managed during hybrid meetings will strengthen the methodology. Are there any fallback plans for managing (potential) panelist dropouts (it is a common challenge for a Delphi), particularly in subgroups? This should be noted.

Grammar and Style

The grammar and style of the paper are clear but could benefit from a careful edit and re-read. Several sentences, particularly in the "Study Record Retention" and "Research Data Management" sections, are lengthy and could be restructured into shorter, more readable forms. Redundancy in describing data encryption and ethical compliance could be streamlined to avoid repetition. And, yes I know this is stylistic, but use of active voice would improve the paper -- especially for professional audiences. I also suggest adding really basic introductory sentences before diving into complex details in sections (like EDI) to improve the overall flow and applicability of content.

A Few Suggestions

To strengthen the draft, maybe provide templates or examples for the demographic questionnaire and controlled feedback forms. These would enhance transparency. Plans for dissemination are good, particularly the integration of a website and social media platforms. Perhaps including strategies for keeping the framework dynamic and updated post-publication would be helpful? Are there strategies for participant fatigue? Such as incentives or even streamlined surveys. Recognizing the potential influence of controlled feedback on panelist opinions and detailing mitigation strategies to minimize bias could improve outcomes.

Overall, the protocol has a good methodology and commitment to inclusivity, but it would benefit from more precise descriptions of statistical methods, panelist selection processes, and importantly an outline of contingency plans. Refining language for consistency, reducing redundancy, and breaking down complex sentences will enhance the clarity and accessibility. It is an applied work, so writing for such audiences is important.

7. PLOS authors have the option to publish the peer review history of their article (what does this mean? ). If published, this will include your full peer review and any attached files.

**Do you want your identity to be public for this peer review?** For information about this choice, including consent withdrawal, please see our Privacy Policy .

Reviewer #1: **Yes: ** Hojatolah Gharaee

Reviewer #2: No

---

## [Author Response · Author response to Decision Letter 1]

13 Feb 2025

Review Comments to the Author

Reviewer #1: Overall, this study protocol is well-prepared and addresses an important issue in public health surveillance. The authors’ structured approach, use of the Delphi method, and comprehensive dissemination plan indicate a strong foundation for a valuable contribution to public health. Nevertheless, to improve the study, the following items can be reviewed and revised:

1. The background and rationale could be strengthened by adding a section specifically addressing the limitations of current evaluation methods and how this framework seeks to address those gaps.

Thank you for providing these insights. We agree with you, and we have introduced a paragraph to address this suggestion.

2. The article could include references to recent advancements or alternative applications of wastewater-based surveillance (e.g., monitoring antimicrobial resistance, drug residues).

We agree with your assessment and have highlighted the role WBS for antimicrobial resistance, with additional statements indicating WBS broad uses.

Consider a pilot round to identify potential issues with the survey questions or consensus-building platform (Surveylet Delphi).

We have incorporated your suggestion in the Procedure section. We will pilot test the survey for up to five panellists, composed of executive group members and other potential panellists matching our recruitment criteria. The pilot test will focus on assessing the clarity and relevance of background information provided for candidate items. The survey will be revised as necessary following feedback from pilot testing.

3. Additional details on post-publication use and integration of the framework in policy-making and public health systems would enhance practical applicability.

We have revised the “Step 5: Dissemination of evaluation framework” section to include further details about our post-publication knowledge translation strategies.

Post-publication, we will aim to keep the framework dynamic and up to date using a website, Discourse server, or other social media platforms to elicit ongoing feedback. Post-publication strategies may also include the creation of guides, toolkits, or ontologies to enhance the practical applicability of the evaluation framework and facilitate its adoption for policy and decision-making.

4. How this framework might adapt to different socio-economic and geographic contexts should be discussed.

Thank you for highlighting the importance of an evaluation framework’s ability to adapt to different socio-economic and geographic contexts. Our Executive Group emphasized the importance of this perspective, and we feel that this is a strength in our methodology, however we now realize that it was under-developed in our manuscript. We have added a description of how we will report e-Delphi findings relating to equity in the “Equity, Diversity, and Inclusion (EDI)” section of the manuscript. We have also included the rating properties for candidate items in the “Defining consensus” section of the manuscript.

Candidate items will be evaluated across four rating properties: (1) Relevance and practical utility; (2) scientific rigor, validity and reliability; (3) feasibility, adaptability and resource implications; and (4) equity, inclusiveness, and mitigation of bias. These rating properties were chosen to ensure that the framework addresses the needs of varied settings. From this, we will assess the level of consensus on these candidate items, specifically highlighting candidate items rated as strong or weak in terms of their equity implications. Of note, after reviewing the literature, our study is the first study that we are aware of that uses equity as a specific rating property.

Where feasible, and with sufficient statistical power, we will further analyze the data by subgroup, distinguishing respondents from high-, middle-, and low-income settings, as well as different world regions. This will allow for a nuanced understanding of how these contexts influence the development of the framework and which candidate items may be most valued in these areas.

5. Almost, it could be say that if any other researcher were to conduct this study, they would likely use the same approach. As a result, I believe there is a lack of specific innovation or creativity in this protocol. It is recommended to consider modifications to the research methodology or interventions to enhance the protocol's appeal and impact, thereby creating more pronounced distinctions from similar studies.

While we appreciate the perspective regarding the potential similarity of approaches among researchers, we respectfully suggest that the strength of this protocol lies in its emphasis on rigour and reproducibility. Using a multinational, multidisciplinary e-Delphi process ensures an evidence-based, comprehensive and collaborative approach. This methodology prioritizes consensus-building among diverse experts. We note that rigorous scoping reviews and consensus studies have not been performed for surveillance evaluation frameworks.

The concept of international consensus for public health policy and decision-making was elevated during the Covid-19 pandemic [1–3]. Therefore, our innovation is in applying lessons learned from consensus-building during the pandemic to this space. Given, that wastewater-based surveillance is multidisciplinary and currently evolving, we feel that is would benefit from a rigorous consensus-based exercise.

Works Cited:

1. Lazarus JV, Romero D, Kopka CJ, Karim SA, Abu-Raddad LJ, Almeida G, et al. A multinational Delphi consensus to end the COVID-19 public health threat. Nature. 2022;611: 332–345. doi:10.1038/s41586-022-05398-2

2. Hillmer MP, Feng P, McLaughlin JR, Murty VK, Sander B, Greenberg A, et al. Ontario’s COVID-19 Modelling Consensus Table: mobilizing scientific expertise to support pandemic response. Can J Public Health. 2021;112: 799–806. doi:10.17269/s41997-021-00559-8

3. Balestracci B, La Regina M, Di Sessa D, Mucci N, Angelone FD, D’Ecclesia A, et al. Patient safety implications of wearing a face mask for prevention in the era of COVID-19 pandemic: a systematic review and consensus recommendations. Intern Emerg Med. 2023;18: 275–296. doi:10.1007/s11739-022-03083-w

Reviewer #2: The methods and protocol sections are comprehensive and well-structured. This reflects a good approach to the Delphi process.

6. I think there is an opportunity to provide additional detail to better support clarity and detail reproducibility. For instance, specifying the scoring scale used by panelists and clearly defining the criteria for achieving consensus would strengthen the methodology.

Thank you. We agree and we have revised to better explain the consensus rating process. We have already incorporated the details of the fully labelled 7-point Likert scale to be used by panellists in Fig 2. However, we have now included the rating properties that will be used to score each candidate item. We have incorporated your feedback in the “Defining consensus” section of the manuscript and are including a new supporting information file as part of the revision titled “S4 Appendix. Rating properties for e-Delphi candidate item scoring.”

7. The protocol emphasizes EDI, but the selection process for panelists could be better explained to outline how diversity in expertise and demographics will be ensured, and importantly how it might shape/improve the outcomes.

Thank you for your feedback. The study working group will monitor the distribution of panellists based on demographic information submitted during active recruitment. Based on the submitted information, the study working group will conduct targeted recruitment of certain cohorts (e.g., discipline subgroup, world region, etc.) if it appears that they are currently underrepresented. Targeted recruitment of specific cohorts will be conducted to ensure a diversity in expertise and demographics among panellists. This information has been incorporated in the “Recruitment of panellists” and “Equity, diversity, and inclusion (EDI)” sections of the manuscript.

8. While the quantitative analysis is detailed, further explanation of how median, range, and IQR will be used to define consensus and stability is needed. Also, elaborating on why Fisher’s exact test was chosen and its threshold for significance would provide a stronger rationale for the approach. I like to see stats choices well defined.

Fisher’s exact test was chosen for its suitability in analyzing small sample sizes and categorical data, as it does not rely on the assumption of large sample sizes required by other tests, such as the chi-squared test. This makes it a good choice for comparing item ratings across subgroups, particularly given the potential variability in panellist numbers across disciplines, income levels, and regions.

Regarding the threshold for significance, we will use a standard alpha level of 0.05, unless multiple comparisons necessitate an adjustment, such as using the Bonferroni correction. The median, range, and IQR will not be used to define consensus but rather to report stability and the differences between groups. They will only be used for descriptive secondary analyses of e-Delphi results. Including this rationale in the manuscript will provide greater transparency and clarity, and we appreciate your suggestion to ensure our statistical choices are well-defined and justified.

9. The ethical considerations part is strong, though additional information on how informed consent will be managed during hybrid meetings will strengthen the methodology.

We have modified the “Step 4: Consensus meeting” section of the manuscript.

Study executive group members will attend the consensus meeting. Therefore, an informed consent process will not be required, as executive group members are co-authors on this study. The consensus meeting will be guided by a Terms of Reference (ToR) document that executive group members agreed upon at the start of the study process. Chatham House Rule will apply to all executive group meetings, including the consensus meeting. The rule will encourage the free expression of ideas without attributing them to an individual or institution.

10. Are there any fallback plans for managing (potential) panelist dropouts (it is a common challenge for a Delphi), particularly in subgroups? This should be noted.

Thank you for highlighting the importance of having a mitigation plan. Our fallback plans for managing potential panellist dropouts include recruitment and retention strategies. We aim to conduct robust recruitment approximately four weeks before the start of the e-Delphi survey. We will also continue recruiting throughout each e-Delphi survey round (approximately 4-6 weeks). We are allowing new panellists to participate in the second round to mitigate our reliance on panellist from the first round. This means panellists can participate in the first or second round only, or both, depending on their preference. However, to help retention efforts, we only offer compensation to panellists who complete all rounds of the e-Delphi survey.

As noted in our response to question #7, the study working group will monitor the distribution of panellists based on demographic information submitted during active recruitment. Based on the submitted information, the study working group will conduct targeted recruitment of certain cohorts (e.g., discipline subgroup, world region, etc.) if it appears that they are currently underrepresented. Targeted recruitment will occur to help mitigate against uneven panellist dropout across cohorts, to ensure a diversity in expertise and demographics among panellists.

11. The grammar and style of the paper are clear but could benefit from a careful edit and re-read. Several sentences, particularly in the "Study Record Retention" and "Research Data Management" sections, are lengthy and could be restructured into shorter, more readable forms. Redundancy in describing data encryption and ethical compliance could be streamlined to avoid repetition. And, yes I know this is stylistic, but use of active voice would improve the paper -- especially for professional audiences. I also suggest adding really basic introductory sentences before diving into complex details in sections (like EDI) to improve the overall flow and applicability of content.

Thank you for your helpful feedback. We have revised and streamlined the "Study Record Retention" and "Research Data Management" sections. We have also added two introductory sentences to the beginning of the “Equity, diversity, and inclusion (EDI)” section. We have reviewed the entire manuscript to improve the overall flow and applicability of the content.

12. To strengthen the draft, maybe provide templates or examples for the demographic questionnaire and controlled feedback forms. These would enhance transparency.

Thank you for your thoughtful suggestion. We have included two new supporting information files as part of the revision titled: “S4 Appendix. Rating properties for e-Delphi candidate item scoring” and “S7 Appendix. Draft e-Delphi panellist screening questionnaire.”

13. Plans for dissemination are good, particularly the integration of a website and social media platforms. Perhaps including strategies for keeping the framework dynamic and updated post-publication would be helpful?

Thank you for your valuable feedback. We have revised the “Step 5: Dissemination of evaluation framework” section to include further details about our post-publication knowledge translation strategies.

Post-publication, we will aim to keep the framework dynamic and up to date using a website, Discourse server, or other social media platforms to elicit ongoing feedback. We are planning on creating a living review from the scoping studies. Those studies have entailed generating an ontology to synthesize concepts. That ontology will be finalized based on the consensus exercise and published.

Post-publication strategies may also include creating guides and toolkits to enhance the practical applicability of the evaluation framework and facilitate its adoption for policy and decision-making.

14. Are there strategies for participant fatigue? Such as incentives or even streamlined surveys. Recognizing the potential influence of controlled feedback on panelist opinions and detailing mitigation strategies to minimize bias could improve outcomes.

Thank you for highlighting the importance of having a mitigation plan. To help retention efforts, we will only compensate panellists who complete all rounds of the e-Delphi survey. We will administer streamlined survey pathways depending on the panellist’s expertise. For example, members of the engaged public will have the option to skip ascertainment performance measurements to allow them to focus on attributes such as communication and partnerships.

Our fallback plans for managing potential panellist dropouts include recruitment and retention strategies. We aim to conduct robust recruitment approximately four weeks before the start of the e-Delphi survey. We will also continue recruiting throughout each e-Delphi survey round (approximately 4-6 weeks). We are allowing new panellists to participate in the second round to mitigate our reliance on panellist dropouts. This means panellists can participate in the first or second round only, or both, depending on their preference.

As noted in our response to question #7, the study working group will monitor the distribution of panellists based on demographic information submitted during active recruitment. Based on the submitted information, the study working group will conduct targeted recruitment of certain cohorts (e.g., discipline subgroup, world region, etc.) if it appears that they are currently underrepresented. Targeted recruitment will occur to help mitigate against uneven panellist dropout across cohorts, to ensure a diversity in expertise and demographics among panellists.

15. Overall, the protocol has a good methodology and commitment to inclusivity, but it would benefit from more precise descriptions of statistical methods, panelist selection processes, and importantly an outline of contingency

---

## [Decision Letter · Decision Letter 1]

22 Apr 2025

Developing an evaluation framework for public health environmental surveillance: Protocol for an international, multidisciplinary Delphi study

PONE-D-24-33538R1

Dear Dr. Manuel,

We’re pleased to inform you that your manuscript has been judged scientifically suitable for publication and will be formally accepted for publication once it meets all outstanding technical requirements.

Kind regards,

Angela Mendes Freitas

Academic Editor

PLOS ONE

Additional Editor Comments (optional):

Reviewers' comments:

Reviewer's Responses to Questions

**Comments to the Author**

1. Does the manuscript provide a valid rationale for the proposed study, with clearly identified and justified research questions?

Reviewer #1: Yes

Reviewer #2: Yes

2. Is the protocol technically sound and planned in a manner that will lead to a meaningful outcome and allow testing the stated hypotheses?

Reviewer #1: Yes

Reviewer #2: Yes

3. Is the methodology feasible and described in sufficient detail to allow the work to be replicable?

Reviewer #1: Yes

Reviewer #2: Yes

4. Have the authors described where all data underlying the findings will be made available when the study is complete?

Reviewer #1: Yes

Reviewer #2: Yes

5. Is the manuscript presented in an intelligible fashion and written in standard English?

Reviewer #1: Yes

Reviewer #2: Yes

6. Review Comments to the Author

You may also provide optional suggestions and comments to authors that they might find helpful in planning their study.

Reviewer #1: I have carefully reviewed the authors' responses to my initial comments and the corresponding revisions made to the manuscript. I appreciate the thorough and thoughtful manner in which the authors have addressed all the raised concerns.

Reviewer #2: The authors have addressed the reviewer comments and suggestion. I look forward to seeing how the project is implemented. It has relevance to other Delhi approaches and will be a helpful; model in other fields.

7. PLOS authors have the option to publish the peer review history of their article (what does this mean? ). If published, this will include your full peer review and any attached files.

**Do you want your identity to be public for this peer review?** For information about this choice, including consent withdrawal, please see our Privacy Policy .

Reviewer #1: No

Reviewer #2: No

---

## [Editor Report · Acceptance letter]

PONE-D-24-33538R1

PLOS ONE

Dear Dr. Manuel,

I'm pleased to inform you that your manuscript has been deemed suitable for publication in PLOS ONE. Congratulations! Your manuscript is now being handed over to our production team.

Kind regards,

on behalf of

Dr. Angela Mendes Freitas

Academic Editor

PLOS ONE